# Perinatal Undernutrition, Metabolic Hormones, and Lung Development

**DOI:** 10.3390/nu11122870

**Published:** 2019-11-23

**Authors:** Juan Fandiño, Laura Toba, Lucas C. González-Matías, Yolanda Diz-Chaves, Federico Mallo

**Affiliations:** Laboratory of Endocrinology (LabEndo), The Biomedical Research Centre (CINBIO), University of Vigo, Campus Universitario de Vigo (CUVI), 36310 Vigo, Spain; jufandinogom@gmail.com (J.F.); lauratoba7@gmail.com (L.T.); lucascgm@uvigo.es (L.C.G.-M.); dizyolanda@gmail.com (Y.D.-C.)

**Keywords:** lung development, undernutrition, lung diseases, ghrelin, leptin, GLP-1, retinoids, cholecalciferol, fetal growth restriction, respiratory distress syndrome

## Abstract

Maternal and perinatal undernutrition affects the lung development of litters and it may produce long-lasting alterations in respiratory health. This can be demonstrated using animal models and epidemiological studies. During pregnancy, maternal diet controls lung development by direct and indirect mechanisms. For sure, food intake and caloric restriction directly influence the whole body maturation and the lung. In addition, the maternal food intake during pregnancy controls mother, placenta, and fetal endocrine systems that regulate nutrient uptake and distribution to the fetus and pulmonary tissue development. There are several hormones involved in metabolic regulations, which may play an essential role in lung development during pregnancy. This review focuses on the effect of metabolic hormones in lung development and in how undernutrition alters the hormonal environment during pregnancy to disrupt normal lung maturation. We explore the role of GLP-1, ghrelin, and leptin, and also retinoids and cholecalciferol as hormones synthetized from diet precursors. Finally, we also address how metabolic hormones altered during pregnancy may affect lung pathophysiology in the adulthood.

## 1. Introduction

Maternal diet is an essential factor that controls fetal growth, both directly by providing nutrients to the embryo and indirectly by regulating the expression of endocrine mechanisms that control the uptake and use of nutrients by the fetus; it also contributes indirectly by changing epigenetic profile and so modulating the expression of genes. The reduction in caloric supply during pregnancy that usually comes accompanied by deficiency of macro and several oligonutrients is called maternal undernutrition. It is demonstrated that maternal undernutrition reduces fetal and placental growth in animals and humans [1]. The reduction in fetal growth is explained by the reduction in cell division [2], which is the result of the adaptation of the cells to the lack of nutrients and the alteration of growth factor and hormone supplies, especially insulin and growth hormone [3]. Fetal growth restriction (FGR) is defined as the fetal growth in lower rate than the normal growth potential, and is an important cause of fetal and neonatal morbidity and mortality [4].

Lung development is a complex process that initiates in utero and continues until early adulthood. In humans, lung development starts as soon as week 3 of gestation [5]. Lung organogenesis comprises five differentiated stages in humans [6]. In the embryonic stage (4 to 6 weeks of gestation, WG), the two lung buds and primary bronchi emerge from the primitive foregut. In the pseudoglandular stage (5 to 17 WG), there is an expansion of the conducting airways. Following this, in the canalicular stage (16 to 27 WG), the epithelia differentiates to separate conducting and respiratory airways and the pulmonary surfactant starts to be synthetized by alveolar type II cells (ATII). In the saccular stage (28 to 31 WG), there is a transition from branching morphogenesis to alveologenesis. In the final alveolar stage (32 WG until early postnatal life), alveoli form and grow.

Other mammal species used for the study of lung development show similar stages, but at different timing during gestation. Rodents have an immature lung at birth—they are in the saccular stage and the alveoli develop postnatally [7]. The deficit of nutrients may alter normal lung development, and promotes a long-lasting impact in the lung structure and function [8].

## 2. Effect of Metabolic Hormones in Lung Development

Hormones and growth factors lead lung morphogenesis. Some key hormones for metabolic control such as insulin, glucocorticoids, and thyroid hormones are at the core of regulatory management of organ development. However, there is extensive literature about their role in lung development and organogenesis that the interested reader might easily find, and thus they are not included in our review, despite their undoubted relevance. 

Instead, new hormones modulating metabolism have been recently shown to have a key role in the maturation of several organs, including the lung. In the next paragraphs, we summarize the actions of some of the most relevant metabolic hormones, such as ghrelin, leptin, GLP-1, and gene-regulating hormones such as retinoids and cholecalciferols. 

### 2.1. Ghrelin

Ghrelin is a 28 amino acid acylated peptide derived from preproghrelin, a 117 AA precursor. It was firstly identified in rat and human stomachs [9], but later, ghrelin expression was found in other adult organs such as the pituitary, hypothalamus, kidneys, heart, and placenta [10]. Ghrelin acts trough a G protein-coupled receptor known as growth hormone secretagogue receptor subtype 1a (GHS-R1a) [9], because it potently stimulates growth hormone (GH) release from the pituitary. In addition, ghrelin stimulates food intake, acting at hypothalamus, and it is involved in the regulation of metabolism, having an overall anabolic effect [11]. 

Ghrelin hormone is detected in cord blood in human fetuses from 20 weeks of gestation [12]. Interestingly, ghrelin is expressed in neuroendocrine cells of the bronchial wall in the pseudoglandular stage of fetal lung development (7–18 WG), but its levels decrease from 19 WG to the second year of postnatal life, and remains afterwards [13]. GHS-R1a is also widely expressed in fetal lung tissue [14,15]. It has been postulated that ghrelin acts as a regulator of fetal lung development in an autocrine/paracrine way, and when exogenously administered, it contributes to fetal lung branching in in vivo and in vitro studies [15,16]. 

It has been observed that in congenital diaphragmatic hernia (CDH), the ghrelin gene is overexpressed in humans and in an animal model of CDH induced by nitrofen administration. These data suggest a potential role of ghrelin in the mechanisms involved in attenuation of lung hypoplasia [16]. In addition, and very relevant, ghrelin administration sensitizes lung fetal tissue to the action of retinoic acid (RA) by upregulating RA receptors, what may be part of the underlying mechanism to explain the effect of ghrelin in lung growth and development [17]. Moreover, ghrelin administration improved pulmonary hypertension and attenuated pulmonary vascular remodeling in newborn pups from an animal model of persistent pulmonary hypertension [18]. 

### 2.2. Leptin

Leptin is a 164 AA peptide product of the *ob* gene [19]. Leptin is produced and secreted by the white adipose tissue, and so it is considered an adipokine. The circulating leptin levels seem to be related to the whole amount of fat stored in adipose tissues [20]. Interestingly, leptin has shown to have pleiotropic effects and it can modulate food intake and energy expenditure, immune response, reproduction, and blood pressure homeostasis [21]. 

Leptin acts through the leptin receptor (Ob-R), encoded by the *db* gene [22]. Ob-R is a member of the class I cytokine receptor family and it is composed for six different isoforms, all of them products of the alternative splicing of the Ob-R mRNA [23]. Leptin and Ob-R are expressed in many other tissues, like the placenta and lungs [24,25]. In lung adult tissue, leptin expression was identified in bronchial epithelial cells, ATII cells, which produce the surfactant, and in alveolar and interstitial macrophages [26]. Leptin receptor expression is detected in the distal lung both in the alveolar and bronchial epithelia [27].

In the fetal lung, leptin gene is expressed in lipofibroblasts, and its levels increases during alveolar differentiation, when pulmonary surfactant phospholipid synthesis is induced [28]. The lung is one of the few tissues that expresses the Ob-R leptin receptor during fetal development [29,30]. Ob-R is expressed specifically by fetal ATII cells [30] and it is enhanced in late gestation, which suggests a key role of leptin in lung maturity [31]. In FGR, the expression of leptin and Ob-R diminishes during the canalicular stage of lung development, being a relevant pathogenic event to explain the lung immaturity in that condition [32]. Moreover, leptin increases surfactant-associated protein (SFTP) expression in in vitro culture of fetal lung explants and in fetal ATII cells [32,33]. The stimulation of SFTP production by leptin is been postulated to be the result of a regulatory paracrine feedback loop between the lipofibroblasts inside the lung and the type II alveolar epithelial cells [28]. On the other hand, the administration of leptin to control animals in vivo produces contradictory findings. While leptin did not modify surfactant synthesis in sheep and mice fetal lungs [34], in more recent studies, it was able to increase the mRNA expression of surfactant-associated protein B (SFTPB) in fetal ewes [35]. In addition, leptin administration to pregnant rats between GD19 and GD20 prevented the alterations in fetal lung architecture and normalized the expression of surfactant-associated protein A (SFTPA) in a model of FGR [32].

Furthermore, the role of leptin in lung maturation may be clarified in *ob/ob* mice, which lack leptin expression [36]. These mice, which show a clear obese phenotype, also present an altered alveolar formation that may be observed from the second week of postnatal life onwards, with a clear decrease in lung volumes and reduced alveolar number and total alveolar surface area. The postnatal leptin replacement in *ob/ob* mice stimulates alveolar enlargement and increases lung volume and alveolar surface area [37]. 

On the other hand, the excessive leptin levels may also have deleterious effects. In rats, a maternal high fat diet increases offspring serum leptin levels, and increases inflammatory cell infiltration and interstitial remodeling, although in this case, it is not clear whether these effects might alternatively be secondary to obese phenotype and dysregulation of metabolism [38]. In fact, there is a negative correlation between leptin levels and forced expiratory volume in first second (FEV1), in obese children and adolescents [39].

Again, all of the little experimental data reported to date clearly indicate that leptin may play a relevant role in lung development, and in some way, this hormone might contribute to explain the functional and pathophysiological connections already observed between adipose tissue and lungs.

### 2.3. GLP-1

Glucagon-like peptide 1 (GLP-1) is an insulinotropic hormone produced by enteroendocrine L-cells of the ileum in response to food intake [40]. GLP-1 is the product of post-translational processing of proglucagon gene. GLP-1 acts by binding to GLP-1 receptor (GLP-1R), a G protein-coupled receptor that is widely expressed in many tissues, including a very high expression in fetal and adult lungs [41,42]. 

During fetal development, GLP-1 receptor is expressed in lung tissue, and its expression is greatly increased just immediately before birth, in coincidence with a period of high surfactant demand before alveolar expansion at first breath after birth [42]. GLP-1R activation increases in vitro phosphatidylcholine secretion in rodent and human ATII cell primary cultures [43,44]. GLP-1 analogue, exendin-4 also, increases perinatal SFTP expression and secretion in rats [42]. In the animal model of lung hypoplasia induced by nitrofen in pregnant rats, exendin-4 administration promotes the expression of SFTPA and SFTPB in similar amounts as dexamethasone, but it also improves the structural development of alveoli and the interstitial tissue, thus allowing the survival of a significant number of newborn rats [42], which never found in untreated animals. In addition, transplacental administration of the GLP-1R agonist liraglutide improved the morphology of the pulmonary vascular vessel in an animal model of congenital diaphragmatic hernia in rabbits [45]. Moreover, GLP-1R activation promoted a marked induction of the ACE2 expression, which enhanced the activity of the ACE2/Ang(1-7)/MasR branch of the renin–angiotensin system, with vasodilatory instead of vasoconstrictor properties, in an animal model of FGR by perinatal food restriction of the mothers [46]. This was also observed in diabetic rats, showing right ventricle hypertrophy, which is prevented by just one-week administration of liraglutide [47].

In summary, our group and others have shown that GLP-1 receptor agonists have very important effects in different aspects of lung physiology. These molecules stimulate the production of both components of surfactant, phospholipids, and SFTPs; they regulate the vascular tone of the pulmonary vessel, promoting vasorelaxation instead of vasoconstriction, thus preventing pulmonary hypertension by the modulation of the components of the renin-angiotensin system; and they improve the alveolar and interstitial histological structure of the lung tissue (Figure 1). GLP-1 receptor expression is also regulated in relation to key events in the lung physiology, and it is overexpressed immediately before birth. Altogether, GLP-1 receptor agonists show protective effects that improve lung function in different physiological and pathophysiological conditions, suggesting a very relevant role in this organ.

### 2.4. Retinoids

Retinoic acid (RA) is a metabolite derived from diet that acts properly like a hormone regulating gene expression [48]. This hormone is obtained as a micronutrient, either as retinyl-esters, present in animal origin food, or as carotenoids, present in vegetables. The hepatocytes are the main reservoir of retinoids, where they accumulate up to 70% in the form of retinyl-esters [49]. When an extrahepatic tissue requires RA, retinyl-esters are cleaved to retinol, which is transported to target tissue bound to retinol-binding protein (RBP) [50]. In target tissues, retinol undergoes two successive oxidations to produce all-trans retinoic acid (ATRA), the biologically active hormone [49]. 

Retinoids exert their actions through two different families of nuclear receptors that are ligand-dependent transcription factors: Retinoid acid receptors (RARs) and retinoid X receptors (RXRs) [51]. ATRA binds to an RAR, which then forms a heterodimer with an RXR molecule. This complex binds to specific retinoic acid response elements (RAREs) present in the genomic DNA upstream of the sequence of the gene promotor region. The RA receptors work as hormone-dependent transcription regulators of several genes, interacting with other hormonal families such as estrogens and thyroid hormones [51].

ATRA plays an essential role in fetal development and in tissue homeostasis by regulating cellular differentiation, tissue maturation, remodeling and apoptosis, and tissue repair [52]. It was shown that reduced levels of RA might promote fetal malformations of several organs [53], whereas very high levels may trigger teratogenesis [54]. Therefore, circulating RA levels must remain within normal ranges during pregnancy.

During fetal lung development, and since early embryonic stages, there is synthesis of RA and expression of RA receptors in the primordial lung buds [55,56]. In fact, RA regulates the formation of the bronchial tubules during the pseudoglandular phase [57]. This may be why the maternal deficiency of retinoid results in lung hypoplasia—and even lung agenesis in the most severe cases [53,58]. This effect was shown in fetuses of the RARα/RARβ2 double knockout mice, which had blunted the capacity to respond to RA [59]. 

In murine animal models, lung maturation is finished after birth. In that case, it can be observed that RA also plays a key role during perinatal lung maturation, when there is a relative depletion of retinyl-ester levels in lung tissue [60], but RAR expression is upregulated in alveoli with respect to mature lung, which suggests RA is involved in the genesis of alveoli [61]. Moreover, RA induces the proliferation and differentiation of fetal type II to type I alveolar epithelial cells in vitro [62], and it increases the expression of mRNA for surfactant-associated protein D [63].

Congenital diaphragmatic hernia (CDH) is a major life threatening disease, characterized by a failure in both alveolar and vascular pulmonary development [64]. There is evidence that a defective mechanism in the retinoid signaling pathway is involved in the etiology of CDH [65]. In classical studies, an incidence of 25–70% of CDH in the offspring of pregnant rats with a deficient intake of RA precursors was reported [66]. Whereas in humans, CDH-affected newborns present a 50% reduction of plasma levels of retinol and retinol binding protein with respect to healthy newborns [67]. 

ATRA has been shown to also be very effective in other animal models of lung diseases. For example, in bronchopulmonary dysplasia (BPD), the pups are exposed to hypoxia conditions from postnatal day 1, disrupting normal septation and lung alveolarization [68]. In this model, postnatal treatment with RA improves alveolar structure, reduces septal fibrosis, and increases survival [69,70,71]. 

In addition, RA contributes to ameliorate the status of the pups in experimental models of lung hypoplasia. In one of these models, the perinatal caloric restriction decreases the RARα expression, and the intraperitoneal administration of RA to the pups improves alveolar formation, likely overcoming the partial deficit of receptors but also stimulating the expression of RARα [72]. 

There are different strategies for modeling lung hypoplasia in laboratory animals. Nitrofen (2, 4-dichlorophenyl-p-nitro phenyl ether) is a molecule developed as a herbicide, without toxicological effects in adult rats. However, administration of nitrofen to pregnant rats on day 9 of gestation induces deep alterations in lung development, to the end that pups show lung hypoplasia, making them not viable for extra uterine life [73]. Nitrofen-induced lung hypoplasia might involve abnormalities in the synthesis, uptake, and signaling pathway of the retinoid system [74,75,76], since retinoid administration to lung explants of nitrofen-treated animals greatly improves the indicators of lung growth [77]. In addition, in vivo, the administration of retinoid precursors to the mothers during gestation reduces the incidence of CDH, and increases survival and lung maturity of the litters [78,79]. It was also shown that RA administration to mothers treated with nitrofen increases postnatal alveologenesis in the progenies [80], likely by promoting the proliferation of type I alveolar epithelial cells [81]. All these data suggest that an increase in the substrate supply for RA synthesis could counteract the decreased activity of the retinal dehydrogenase 2 (RALDH2), a key enzyme in retinol synthesis, as observed in nitrofen-treated fetal lungs [74]. 

### 2.5. Cholecalciferol

Cholecalciferol or vitamin D3 is a secosteroid that is obtained directly from food of animal origin, or indirectly by synthesis in the skin from 7-dehydrocholesterol after ultraviolet B exposure. This prohormone is inactive, and it experiences two sequential hydroxylation steps to produce 1,25-Hydroxyvitamin D (1,25(OH)_2_D), the hormone active form [82]. 1,2(OH)_2_D, also called calcitriol, interacts with this specific receptor, called vitamin D receptor (VDR), that is a ligand-dependent transcription factor [83]. After ligand binding, it requires the formation of a heterodimer with a retinoid X receptor (RXR) to interact with vitamin D response elements (VDRE) present in the DNA and to regulate gene expression [84]. 

As the action mechanism of calcitriol requires forming heterodimers with the promiscuous receptor for retinoids (RXR), it is not surprising that it may be involved in lung development, maturation, and functional regulation. In fact, the lung is likely one of the main target tissues for calcitriol during fetal development [85]. VDR receptor is expressed in fetal ATII cells, where its activation induces proliferation and the synthesis and secretion of surfactant of both the fractions proteins and phospholipids [86,87,88,89]. The incubation of human fetal and adult ATII cells with 1,25(OH)_2_D in vitro culture increases VDR and the expression of SFTPB [90].

The maternal calcitriol deficiency during lung development in animal models modifies the expression of genes involved in organ development, branching morphogenesis, and regulation of inflammation process [91]. Therefore, several respiratory parameters may be affected, including the reduction in lung volume, vital capacity and oxygen saturation, and increases in airway smooth muscle mass and airway contractility [92,93,94]. All of these changes alter normal lung physiology and might compromise the survival of the litters. In this condition, the supplementation of mothers with calcitriol precursors completely prevents the negative effects of the deficiency. In addition, calcitriol supplementation during lactation in rodents with previous deficiency during gestation improves alveolar septation and lung function [95]. Even in pups from normal pregnant rats, the aerosol administration of calcitriol precursors contributes to lung maturity by increasing the expression of markers of epithelial, mesenchymal, and vascular differentiation that is followed by an increase in the synthesis of surfactant phospholipids [96]. In human studies, there is an association between a reduction of calcitriol levels in 18 WG and a reduced lung function in childhood [97]. It is also demonstrated that severe deficiency of 25(OH)D in preterm infants is related to the development of respiratory distress syndrome [98]. Thus, preterm supplementation with calcitriol precursors reduces the time of assisted ventilation and oxygen supplementation [99], which confirms the essential role of this hormone in lung maturation. In addition, it has been proposed that supplementation with calcitriol precursors during pregnancy may be an effective mean of preventing childhood asthma [100].

## 3. Effect of Undernutrition on Lung Development and Adult Lung Function

There are several different animal models for the study of FGR, including genetic manipulation models, but also mother food restriction during pregnancy [101]. The most frequently used animal species for modeling FGR are mice, rats, and lambs. 

In a model of lamb FGR by the removal of endometrial caruncles, there is a reduction in fetal lung weight, lung liquid volume, and phospholipid concentration in liquid of alveolar lavage [102]. In this model, the lung weight is reduced by a similar rate to fetal body weight reduction, but carrying structural alterations that reveal a retarded maturation [103]. FGR reduces alveolar number and vascular density, but increases septal thickness [104,105]. These alterations become more pronounced during postnatal lung development [104], which leads to a smaller number of large alveoli, alveolar fenestrations, and increased number of mast cells in the lungs of adult animals, anticipating a premature lung aging [106]. At least part of these changes in lung architecture could be explained by a marked reduction in elastin synthesis and deposition [107]. FGR also promotes the reduction of the mRNA and protein expression of the SFTPs [108]. SFTP expression is higher after the delivery in FGR ewes due to the activation of the hypoxia-signaling pathway by increasing HIF-2α mRNA expression [109].

FGR alters normal structure of the lamellar bodies of ATII cells involved in surfactant synthesis and secretion, in the saccular stage before birth in rodents [110]. This alteration also reduces mRNA expression of SFTPs [111]. However, after birth, there is a reduction in lung surfactant lipid levels, just in the early postnatal period, without modifying the expression of surfactant-associated proteins in the remaining postnatal period [112]. As described in lambs, FGR also disrupts normal lung architecture in rodents, and it decreases alveolar number and increases septal thickness [113] from postnatal day 1 through adulthood. Moreover, this is accompanied by a decline in synthesis and secretion of elastin, and an increase in static lung compliance [114].

In humans, fetal undernutrition can be caused by at least five situations: (1)Severe nausea and vomiting period that persists more than the first trimester [115];(2)The “Maternal Depletion Syndrome,” a product of a short inter-pregnancy interval, not allowing sufficient time to replenish energy reserves and recovery of mothers, which promotes a depletion of both macro- and micronutrients [116];(3)Teenager pregnancy, where the mother, who may still be growing, competes with the fetus for resources [117];(4)Use and abuse of tobacco [118]; and(5)Alcohol/drugs [119], which may promote placenta under-function and reduced nutrient supply to fetus and/or maternal undernutrition.

There are few studies linking FGR, fetal lung development, and neonatal lung pathology in humans. In fact, there are some conflicting results about the effect of FGR over respiratory distress syndrome (RDS). Several studies have concluded that FGR reduces the incidence of RDS and increases the ratio of lecithin/sphingomyelin in amniotic fluid, a marker of lung maturation [120,121]. They explain this accelerated lung maturation as a consequence of the chronic intrauterine stress that increases fetal glucocorticoid levels. Nevertheless, other studies have concluded that FGR increases the risk of developing RDS and the risk of respiratory failure and death [122], and yet others did not find this relation [123]. On the other hand, there is an association between perinatal growth restriction and an increased risk of developing bronchopulmonary dysplasia in preterm infants [124]. Moreover, low birth weight, but not prematurity, decreases lung size and bronchial airflow, and conversely increases bronchial hyperreactivity in children [125].

In the mature lung, there is a clear relationship between the early fetal nutritional environment and adult pulmonary diseases—despite the mechanistic basis of this relationship being unknown [126]. In the adult lung, there is a suggestive, not fully consistent, association between FGR and pulmonary function in adulthood [127]. There are some evidences that FGR can decrease adult lung function [128], whereas other studies did not find any effect over lung function [129]. Another study shows that prenatal exposure to famine did not modify the lung function, but increased the prevalence of COPD [130]. This risk is greater when severe famine exposure occurs during infancy [131]. Asthma is another lung pathology that is related with FGR. There are some studies that link FGR with an increased risk of developing adult asthma [132], whereas other studies conclude that environmental factors during childhood rather than fetal undernutrition are responsible for the increased risk of developing asthma in adult life [133].

## 4. Undernutrition and Hormones in Lung Development

Undernutrition in pregnancy promotes several changes in metabolic control and hormone levels, which are needed to adapt the energy demands to reduced supplies. It is easy to link a caloric deficit with reduced availability of precursor for hormones that are obtained in diet, such as retinoids and carotenoids [134,135]. However, these precursors may be stored in some amounts in the liver and fat depots. In such a way, nutritional deficits of these hormones must be set up likely before pregnancy, for reducing the reserves enough to affect fetus development during gestation. In developed countries, the follow-up of every pregnant women and nutritional advice should be enough to prevent this kind of deficit. A large part of the population is in the lower range or outside the normal range for cholecalciferols (VitD), which may be especially critical in some susceptible populations: Low sun exposure, low intake of fish and dairy products, obesity, or undernutrition. 

The effect in the modulation on gene transcription by the activation of the retinoid hormone system is so important that it might be a source of teratogeny when in elevated levels during pregnancy. In addition, on the other hand, a deficit of retinoids promotes alterations in reproduction, placentation, and organ development. However, there is not a recommendation to supplement nutrition with retinoid precursors in pregnancy apart from in known deficient populations. In some African countries, this deficit may be present in the 21–48% of all pregnant women [135]. On the other hand, some hormones are involved in the short-term availability of energy resources, and may eventually be relevant in the case of reduced food intake during pregnancy. In this context, and as described above, leptin seems to be a relevant hormone in lung development. This hormone is mainly secreted by adipose tissue in proportion to total fat storage. During starving, even partial, fat depots and, consequently, leptin circulating levels are reduced [136]. Leptin is also produced by the placenta, where it plays a local role in protein synthesis and proliferation of placental cells. It has been also postulated that leptin is very important for maternal–fetal exchanges, regulating the growth and development of many organs, including the lung. In fact, dysregulation of leptin mechanisms is link to several disorders occurring in pregnancy, such as gestational diabetes and intrauterine growth restriction [137]. In FGR neonates, there is a reduction in circulating leptin levels, due to a reduction in fetal fat mass and placental production [138,139] The fetal reduction in leptin levels may compromise correct lung development. The reduction in fetal circulating leptin levels is usually compensated by a postnatal increase when enough energy supply is set up, which explains the catch-up lung growth in FGR offspring [140]; however, it may also be related to the augmented incidence of childhood asthma in FGR offspring [141].

Another hormone that has a relevant role in metabolic and food intake control is ghrelin. Ghrelin is a peptide with orexigenic, adipogenic, and GH-releasing properties [142]. Regarding all described effects for ghrelin, it is important in the regulation of metabolism and it has been suggested that it contributes to energy resource distribution, linking nutrients to growth and development of the organs. Ghrelin levels vary during pregnancy, reaching the highest peak at mid-gestation, and then declining up to term [143]. Ghrelin is present in the cord blood and inversely correlates with fetal growth. Moreover, intrauterine ghrelin levels have been linked to programming body weight in the postnatal period [144]. FGR fetuses present high ghrelin levels in response to intrauterine malnutrition, which might contribute to increase neonate appetite, which suggests a role of ghrelin in catch-up growth [145,146]. Nevertheless, more recently, others have shown that ghrelin levels are reduced in “small for gestational age” fetuses [146], and this is in accordance with increased levels of cortisol in FGR fetuses due to the stress in the intrauterine environment. It has been shown that there is a negative correlation between cortisol and ghrelin levels [147]. Despite there being few studies about ghrelin’s involvement in lung function and development, the reported results suggest it has a relevant role. The action mechanisms underlying the effects of ghrelin in the lungs will need some more studies to be revealed. 

GLP-1 is the least studied metabolic hormone, here presented in relation with pregnancy. GLP-1 could compensate pregnancy-related alterations in metabolism, such as an increase in glycaemia and the development of insulin resistance, based on the increase of fasted active GLP-1 levels in the third trimester of gestation [148]. This increase in GLP-1 secretion is a product of gastrointestinal tissue expansion, rather than satiety [149]. GLP-1 circulating levels are reduced in pregnant mothers with gestational diabetes [150,151]. However, we have no data about changes in GLP-1 levels during normal pregnancy. It is important to emphasize that GLP-1 half-life is very short, lower than 2 min. Therefore, GLP-1 levels may change very fast after meals, and so to study GLP-1 variations will demand to do repeated short-interval blood sampling in every individual. In a recent study, it has been reported that GLP-1 and GIP circulating levels in mothers and cord blood negatively correlate with 25OHD, and, surprisingly, GLP-1, GIP, and ghrelin positively correlate with glycated albumin maternal/cord ratio, highlighting the relevance of these hormones and their interplay in the complex control of metabolism, especially in pregnancy.

Ghrelin and GLP-1 are secreted in relation to meals and, since they may serve as a link between maternal food intake and metabolism, may possibly modulate the exchange of nutrients through the placenta. However, and as described above, both hormones have direct and important effects in lung development. It must be highlighted that GLP-1 modulates many different functions of the lung, including key processes such as the production of surfactant components, or the modulation of vascular tone of pulmonary vessels by controlling the renin–angiotensin system local activation. In addition, it should of the greatest interest to study whether the placenta, as the maternal/fetal interchange organ, is a target for GLP-1 modulatory actions, as we have no data in this respect. 

Finally, clinicians dedicated to pregnancy must be conscious of the delay in lung maturity in all of the five clinical situations mentioned above, which include: Persistent severe vomiting beyond first trimester; “Maternal Depletion Syndrome”, especially in susceptible populations; teenager pregnancy; use of tobacco and abuse of alcohol and drugs [119]; but also in obese and diabetic mothers. In all of these cases, a complete, well balanced, and eventually supplemented diet of mothers will guarantee normal lung development of fetuses and newborns, contributing to prevent lung pathology in infancy and adult life. This diet should provide enough, but not an excessive amount of, calories and calcitriol and retinoid sources, in addition to other known nutrients needed for organogenesis, such as good quality protein, iodine, and iron. Although, correct attention to the diet of pregnant women is included in current gestational protocols in occidental medicine, it appears that this is not so general in many countries, and thus should be regarded as a priority objective of preventive health policies.

In conclusion, the reduction of food intake during pregnancy may not just directly affect tissue development because insufficient resources, but also undernutrition modifies the hormonal milieu, which is critical for many organs, including lung. Retinol and cholecalciferol are hormones synthetized from precursors obtained from diet; therefore, reductions in food intake limit the availability of these hormones. In fact, the deficit in cholecalciferol is one of the most frequent in pregnancy, especially in susceptible populations. Gestational undernutrition also reduces fat storage, as well as leptin circulating levels in the medium-term; and daily-reduced caloric intake may affect the levels of hormones regulated in the short-term, linked to meals such as ghrelin and GLP-1. The mentioned hormones have key roles in lung development and maturity, including morphogenesis and structure development, cell proliferation and apoptosis, and many functional processes such as production of surfactant components, activity of the local renin–angiotensin system, and vascular tone of pulmonary vessels (see Table 1). Moreover, undernutrition in pregnancy affects all of these hormonal systems at once, in addition to others also relevant such as insulin and IGFs, thyroid hormones, and glucocorticoids. Therefore, the correction of known specific deficits with diet supplementation during gestation is mandatory and should be included in clinical protocols. The disruption of the hormonal environment during pregnancy becomes especially important when the mothers present metabolic diseases such as diabetes and obesity, despite that caloric intake may be preserved. The dysregulation in hormonal control in altered metabolism in mothers may affect lung development and maturity of the fetus to different degrees, also conditioning higher risk to lung pathology in adult life. In this case, the correction during pregnancy of diet and food intake, in proper amounts and composition, is so important to lung development, like it might be in caloric restriction and undernutrition.

## Figures and Tables

**Figure 1 nutrients-11-02870-f001:**
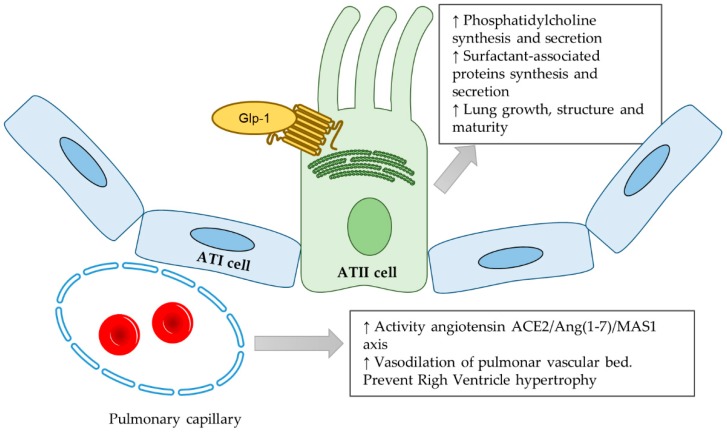
Schematic representation of the main effects of GLP-1R activation in fetal lung tissue. Abbreviations: ATI cell, alveolar type I cell; ATII cell, alveolar type II cell; Glp-1, glucagon-like peptide-1; ACE2, angiotensin-converting enzyme 2; Ang (1-7), angiotensin 1-7; MAS1, Mas proto-oncogene, G protein-coupled receptor.

**Table 1 nutrients-11-02870-t001:** Summary of the effects of the different hormones over lung development.

Hormone	Action in Lung Development	References
Ghrelin	Fetal lung branching	[15,16]
	Upregulating RA receptors/ sensitizing RA action	[17]
Leptin	Enhance lung maturity	[28,31,32,33,34]
	In vitro phosphatidylcholine secretion	[28]
	In vitro SFTPs expression	[28,32,33,34]
GLP-1	In vitro phosphatidylcholine secretion	[43,44]
	In vivo SFTPs expression	[42,46,47]
	Increase ACE2/Ang (1-7)/MasR branch of the renin-angiotensin system	[46,47]
Retinoic acid	Formation of bronchial tubules during pseudoglandular phase	[57]
	Lung maturation	[62,69,70,72,77,78,79]
	In vitro Proliferation of ATII cells and differentiation to ATI cells	[62,81]
	In vitro and in vivo SFTPs expression	[62,63]
Cholecalciferol	Branching morphogenesis	[91]
	In vitro proliferation of ATII cells	[96]
	In vitro surfactant phospholipids secretion	[96]
	In vitro SFTPs expression	[96]
	Lung maturation	[95,96]

Abbreviations: RA, retinoic acid; SFTPs, surfactant-associated proteins; ACE2, angiotensin-converting enzyme 2; Ang (1-7), angiotensin 1-7; MAS1, Mas proto-oncogene, G protein-coupled receptor; ATII cells, alveolar type II cells; ATI cells, alveolar type I cells.

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
