# Peer review of "Perinatal Undernutrition, Metabolic Hormones, and Lung Development"

_nutrients, 2019, doi:10.3390/nu11122870_

Round 1

Reviewer 1 Report

This review was generally well written and structured and focus an important issue. However, in my opinion the paper lacks English coherence, formatting and a conclusion that can improve the readout.

I made the following suggestions:

I suggest adding a definition of maternal undernutrition in Introduction Lines 20-21: I would replace “We will study the role of…” by “We will explore the role of…” since this paper is a review and not an original article Keywords: I would add GLP-1 as one of the keywords since you describe this molecule in the same way you describe Ghrelin, Leptin, Retinoids, … Line 39: I suggest replacing “that regulate” by “that control”, to avoiding repeat the word Line 47: I suggest putting “in utero” in italic Line 78: At the end of the sentence, I would remove “humans” because you wrote "in human foetuses" Line 114: I would replace “to the result…” by “to be the result…” Line 119: It seems to me that there is an additional space between the reference 36 and the beginning of the new sentence Lines 131-134: The new sentence is quite confused. It requires a revision of English in order to bring significance Line 140: The reference 41 is old. Can you provide a more recent one? Line 142: I recommend replacing “&” by “and” Line 146: I recommend putting “in vitro” in italic type Lines 149-150: The sentence has got verbs in the present and in the past. Please reformulate the sentence putting all in the same verb tense Line 159: Put Liraglutide in capital letter as in Line 153 Line 160: Please reformulate the beginning of the sentence since English is not correct Title Figure 1: Please put effects as one word (not effect s), replace “&” by “and”, add legend of abbreviations Line 184: I do not recommend starting a sentence by “And” Line 229: Please remove “(Sugimoto2008)”, as it seems a mistake produced by your reference’s library Line 251: Please replace “however” by other linking word as it does not fit the sentence Table 1: Please pay attention to the following - put a full stop in the title; the title needs to be closer to the table; add legend of abbreviations, put “in vitro” and “in vivo” in italic type Line 310-311: Replace “In the other hand” by “On the other hand” Line 319: Replace “Other study…” by “Another study…” Please confirm the space between lines 325 and 326 Line 329: Please replace “con reduced” - “with reduced” Line 337: Please replace “system are” by “system is” Line 338: Please replace “maybe a source” by “might be a source” Line 357: Replace “Other hormone…” by “Another hormone…” Line 359: Replace “suggested it…” by “suggested that…” Lines 364-369: Put Ghrelin in capital letter as in Line 357 Line 366: English correction “lower small” Line 368: Please put a full stop after “environment” Line 368: I recommend putting a “that” as a linking word between “It has been shown” and “there is a negative” Lines 369-371: Please reformulate this sentence since it is confused Lines 389-381: Please reformulate this sentence since it is confused Line 381: Please rewrite the sentence since English is not correct and I recommend replacing “&” by “and” Line 385: Please add a full stop at the end of the sentence Lines 386-387: Please reformulate Please reformulate this sentence since it is confused Line 387: Please replace “describe” by “described” Line 390-392: Please rewrite the sentence since English is not correct and it does not make sense Throughout the study, the word foetus appears as foetus or fetus. I highly recommend uniformize the word. Please find below where I found out the word fetus instead of foetus:

Lines:16, 24, 45, 46, 88, 106, 108, title figure 1, 187, 189, 231, Table 1, 276, 315, 324, 349, 352, 353, 362, 366

Please add a brief conclusion as a summary

Given these points the manuscript requires minor revisions.

Author Response

This review was generally well written and structured and focus an important issue. However, in my opinion the paper lacks English coherence, formatting and a conclusion that can improve the readout.

I made the following suggestions:

We really appreciate all suggestions made by the referee that greatly improved several aspects of our review.

1-I suggest adding a definition of maternal undernutrition in Introduction.

It has been included a definition of maternal undernutrition following the suggestion of the referee

2-Lines 20-21: I would replace “We will study the role of…” by “We will explore the role of…” since this paper is a review and not an original article.

It has been changed accordingly.

3-Keywords: I would add GLP-1 as one of the keywords since you describe this molecule in the same way you describe Ghrelin, Leptin, Retinoids, …

GLP-1 has been added to the keywords of the manuscript.

4-Line 39: I suggest replacing “that regulate” by “that control”, to avoiding repeat the word.

We agree with this suggestion, and the text has been substituted as demanded.

5-Line 47: I suggest putting “in utero” in italic.

As suggested, “in utero” is now highlighted in italics.

6-Line 78: At the end of the sentence, I would remove “humans” because you wrote "in human foetuses".

We have removed “humans” as suggested, because it is redundant.

7-Line 114: I would replace “to the result…” by “to be the result…”

It is been changed in agreement.

8-Line 119: It seems to me that there is an additional space between the reference 36 and the beginning of the new sentence.

The extra space was suppressed.

9-Lines 131-134: The new sentence is quite confused. It requires a revision of English in order to bring significance.

This sentence has been abbreviated and clarified.

10-Line 140: The reference 41 is old. Can you provide a more recent one?

The cited reference is one of the first work going deep in the regulation of GLP-1 by nutrients. However, in order to a better understanding of the GLP-1 pleiotropic effects, we have substituted it by a more recent review on GLP-1.

11-Line 142: I recommend replacing “&” by “and”.

We have replaced “&” by “and” throughout manuscript.

12-Line 146: I recommend putting “in vitro” in italic type.

We have changed it following the suggestion.

13-Lines 149-150: The sentence has got verbs in the present and in the past. Please reformulate the sentence putting all in the same verb tense.

The sentence is now written in present.

14-Line 159: Put Liraglutide in capital letter as in Line 153.

We really appreciate this comment, however, liraglutide is the name of a molecule, and accordingly it should be better written in lowercases, as it is now all along the text.

15-Line 160: Please reformulate the beginning of the sentence since English is not correct.

We have wrote again this sentence as “In summary, our group and others groups…”

16-Title Figure 1: Please put effects as one word (not effect s), replace “&” by “and”, add legend of abbreviations.

The extra space inside “effects” was suppressed, and “&” substituted as suggested. Finally, we have added all the abbreviations of the cited words in the text of the legend.

17-Line 184: I do not recommend starting a sentence by “And”.

In agreement with the referee´s suggestion we have suppressed “and” at the beginning of the sentence.

18-Line 229: Please remove “(Sugimoto2008)”, as it seems a mistake produced by your reference’s library.

Certainly, it was a mistake, already corrected.

19-Line 251: Please replace “however” by other linking word as it does not fit the sentence.

We have written a clearer sentence.

20-Table 1: Please pay attention to the following - put a full stop in the title; the title needs to be closer to the table; add legend of abbreviations, put “in vitro” and “in vivo” in italic type.

We have addressed all the suggested changes.

21-Line 310-311: Replace “In the other hand” by “On the other hand”.

It has been corrected.

22-Line 319: Replace “Other study…” by “Another study…”

Corrected.

23-Please confirm the space between lines 325 and 326.

Suppressed the extra space.

24-Line 329: Please replace “con reduced” - “with reduced”.

Corrected.

25-Line 337: Please replace “system are” by “system is”.

Corrected.

26-Line 338: Please replace “maybe a source” by “might be a source”.

Corrected as suggested.

27-Line 357: Replace “Other hormone…” by “Another hormone…”

Replaced.

28-Line 359: Replace “suggested it…” by “suggested that…”

Replaced.

29-Lines 364-369: Put Ghrelin in capital letter as in Line 357.

As explained above, “ghrelin” is the name of a molecule and accordingly it must be written in lowercases.

30-Line 366: English correction “lower small”.

We have changed “lower” for “reduced”, in order to clarify the sentence. However, “small for gestational age”, or SGA foetus, is the technical name of a specific condition that we cannot be avoided. We highlighted that condition by quotation marks.

31-Line 368: Please put a full stop after “environment”.

We have corrected this.

32-Line 368: I recommend putting a “that” as a linking word between “It has been shown” and “there is a negative”.

It has been included.

33-Lines 369-371: Please reformulate this sentence since it is confused.

It has been fully changed.

34-Lines 389-391: Please reformulate this sentence since it is confused.

It was been reformulated.

35-Line 381: Please rewrite the sentence since English is not correct and I recommend replacing “&” by “and”.

We have replaced this.

36-Line 385: Please add a full stop at the end of the sentence.

We have added it.

37-Lines 386-387: Please reformulate Please reformulate this sentence since it is confused.

This sentence was completely reformulated.

38-Line 387: Please replace “describe” by “described”.

We have replaced this.

39-Line 390-392: Please rewrite the sentence since English is not correct and it does not make sense.

It was written again in order to be clearer.

40-Throughout the study, the word foetus appears as foetus or fetus. I highly recommend uniformize the word. Please find below where I found out the word fetus instead of foetus:

Lines:16, 24, 45, 46, 88, 106, 108, title figure 1, 187, 189, 231, Table 1, 276, 315, 324, 349, 352, 353, 362, 366

It has been corrected all along the text, and standardized as fetus, as being the most recommended term for scientific purposes.

41-Please add a brief conclusion as a summary

We have included a paragraph as brief conclusion at the end of the paper.

Reviewer 2 Report

Interesting review article. Well written. The authors described how perinatal malnutrition influences lung development. The primary objective of this article is to describe the effect of the derangement of the metabolic hormone on fetal lung development.

I would recommend adding a paragraph for the clinicians about the index of suspicion, diagnosis and management plan in case of suspected cases. Also, I would like to add current recommendations about preventive strategies.

Author Response

Interesting review article. Well written. The authors described how perinatal malnutrition influences lung development. The primary objective of this article is to describe the effect of the derangement of the metabolic hormone on fetal lung development.

I would recommend adding a paragraph for the clinicians about the index of suspicion, diagnosis and management plan in case of suspected cases. Also, I would like to add current recommendations about preventive strategies.

We greatly appreciate the so kind comments of the referee.

Following her/his suggestions, we have included a paragraph about clinical questions at the end of the manuscript, summarizing some general recommendations. However, we would like to emphasize that our review is not focus to stablish specific protocols and recommendations to the clinical practice, what, in our modest opinion, should deserve a separated and specifically addressed paper. Frequent references to results obtained in humans are discussed in the different sections of the review, yielding some recommendations that might be up taken by the reader.  
